# Application of the Extracts of *Punica granatum* in Oral Cancer: Scoping Review

**DOI:** 10.3390/dj10120234

**Published:** 2022-12-09

**Authors:** Mario Dioguardi, Andrea Ballini, Diego Sovereto, Francesca Spirito, Angela Pia Cazzolla, Riccardo Aiuto, Vito Crincoli, Giorgia Apollonia Caloro, Lorenzo Lo Muzio

**Affiliations:** 1Department of Clinical and Experimental Medicine, University of Foggia, Via Rovelli 50, 71122 Foggia, Italy; 2Department of Precision Medicine, University of Campania “Luigi Vanvitelli”, 80138 Naples, Italy; 3Department of Biomedical, Surgical, and Dental Science, University of Milan, 20122 Milan, Italy; 4Department of Basic Medical Sciences, Neurosciences and Sensory Organs, Division of Complex Operating Unit of Dentistry, “Aldo Moro” University of Bari, Piazza G. Cesare 11, 70124 Bari, Italy; 5Unità Operativa Nefrologia e Dialisi, Presidio Ospedaliero Scorrano, ASL (Azienda Sanitaria Locale) Lecce, Via Giuseppina Delli Ponti, 73020 Scorrano, Italy

**Keywords:** oral cancer, *Punica granatum*, pomegranate, OSCC

## Abstract

The *Punica granatum* L. is an ancient fruit plant native to south-western Asia. It belongs to the Litraceae family and of its genus we have only one other *Punica protopunic* species. The fruit is rich in polyphenols, whose extract is consumed as a food and is considered safe. In medicine, it is used for its antioxidant properties; it has a rich component of tannic polyphenols among which the most bioactive are: punicalagin (flavonoids) and anthocyanins (delphinidin, cyanidin, pelargonidin), which are found mainly in the skins and in the pericarp; however, all the parts of the *Punica granatum* are used for therapeutic purposes as anti-inflammatories and analgesics and in diabetes and cardio-vascular disease. *Punica granatum* extracts also show interesting anticancer activities in influencing tumorgenesis and angiogenesis and cell transformation and proliferation. The purpose of this scoping review is to summarize all the scientific evidence on the possible applications of *Punica granatum* extracts in the treatment and prevention of oral cavity tumors to investigate the anticancer properties of the active ingredients extracted from *Punica granatum.* Methods: The scoping review was carried out following the PRISMA-ScR checklist; the search was performed on three databases (Scopus, Science direct and PubMed) and one registry (Cochrane library). Results: The search produced a number of bibliographic sources equal to 11,403; with the removal of duplicates, 670 potentially admissible articles were obtained, from 24 of which only 7 in vitro studies on OSCC cell lines were included. Conclusions: From the preliminary data on the cellular lines of OSCC, it emerges that for oral cancer there are conditions for which the extracts of *Punic granatum* are effective at least from a prevention perspective.

## 1. Introduction

The Pomegranate (*Punica granatum* L.) (Figure 1) is an ancient fruit plant native to south-western Asia (Iran, Pakistan, Afghanistan) but, by virtue of its wide adaptability to pedoclimatic conditions, the plant has been widely cultivated in the rest of the world; it belongs to the Litraceae family and of its genus we have only one other species, the *Punica protopunica*. There are about 500 cultivars in the world that differ in their morphological and nutritional characteristics; the fruit is rich in polyphenols whose extract is consumed as food and is considered safe [1].

In medicine, it is used for its antioxidant properties; it has a rich component of tannin polyphenols among which the most bioactive are: punicalagin (flavonoids) and anthocyanins (delphinidin, cyanidin, pelargonidin), which are mainly found in the skins and in the pericarp, but also in the arils, flowers and leaves (Figure 2); this small tree is therefore rich in bioactive molecules with therapeutic activity [2].

However, all the parts of the *Punica granatum* have been used for therapeutic purposes: leaves, flowers, bark, roots, peel, juice, seeds; their uses find application as anti-inflammatories [3], analgesics [4], in the management of postmenopausal symptoms, being weakly estrogenic [5], but also in erectile dysfunction, in diabetes [6,7,8] and cardiovascular disease [9], in the treatment of arthritis [10] and in Alzheimer’s disease [11]; it also has bactericidal [12], antifungal [13] and antiviral properties [14]; its use in chronic gingivitis has also been evaluated in the dental field [15,16,17,18,19].

*Punica granatum* extracts also show interesting anticancer activities in influencing tumorgenesis and angiogenesis and cell transformation and proliferation [20]. *Punica granatum* has therefore been tested for use in the prevention of many cancers [21].

Its ability to inhibit cell proliferation has been tested on many cancer cell lines such as breast [22], leukemia [23], prostate [24], lung [25] and fibrosarcoma [26] but its antitumor effect on oral cancer cell lines has not been adequately documented; there are only a few studies in the literature that have investigated the anticarcinogenic role of *Punica granatum* in the cell lines of squamous cell carcinoma [27].

Among the main oral cancers, we have oral squamous cell carcinoma (OSCC), which represents about 90% of total cases [28]; histologically, it consists of a multi-layered epithelium consisting of squamous cells with the presence of cellular and nuclear atypia [29]; clinically, it can manifest itself as an ulcer (erythroplakia) [30], or a white plaque (leukoplakia) [31,32] sometimes ulcerated with poorly defined margins, the localization of which can affect the entire epithelium of the oral cavity [31]. The annual incidence in the world is 350,000 new cases diagnosed, with a higher incidence in the population over 50 years of age [33].

Five-year survival is less than 50% [34] and the therapies used to cope with the cancer lead to a moderate reduction in quality of life with difficulties in speaking, swallowing, feeding and with difficulties for patients in relational life [35].

The main recognized risk factors are smoking and alcohol with a synergistic effect; moreover, traumatic events can contribute to the pathogenesis of OSCC, along with genetic factors [36,37], ultraviolet radiation, immunosuppression and nutritional factors.

Precisely in order to be able to reduce the risk factors for OSCC through the use of compounds with high nutritional value, the research has turned its attention to traditional natural medicine and to the identification of foods that could contain bioactive principles towards pre-cancers and cancers; *Punica granatum* is therefore an ideal candidate for its high content of polyphenols, tannins and flavonoids [38]. In this scoping review, we will focus on identifying those studies that investigated the effects of *Punica granatum* extracts on oral cancer and more specifically on OSCCs, summarizing the main results and the state of the research at the present time

## 2. Materials and Methods

### 2.1. Protocol and Registration

The scope review was written and performed following the PRISMA-ScR (PRISMA Extension for Scoping Reviews) checklist as reported by Tricco et al. [20]; the scoping review protocol was registered prior to its execution on INPLASY (the International Platform of Registered Systematic Review and Meta-analysis Protocols), with registration numbers INPLASY 202290027 and 10.37766/inplasy2022.9.0027. 

### 2.2. Eligibility Criteria

All studies investigating *Punica granatum L.* in association with oral and precancerous cancer were considered potentially admissible; no restrictions were applied in relation to the year of publication and based on the language provided that an abstract in English was available. Literature reviews were excluded and were used only as sources for bibliographic research.

### 2.3. Information Sources

The search was carried out on three databases (PubMed, Scopus and Science Direct) and a register (Cochrane library); in addition, a gray literature search was performed on Google scholar and Opengray (DANS EASY Archive); potentially eligible articles were also searched among references from literature reviews on *Punica granatum L.*

The research was conducted between 1 August 2022 and 15 August 2022, with a last update of the records identified on 19 August 2022.

### 2.4. Search

The authors responsible for researching the studies used the following key words in the databases: punica granatum AND cancer. The key words used on PubMed are shown below; Search: punica granatum AND cancer (“pomegranate” (MeSH Terms)) OR “pomegranate” (All Fields) OR (“punica” (All Fields)) AND “granatum” (All Fields) OR “punica granatum” (All Fields) AND (“cancer s” (All Fields)) OR “cancerated” (All Fields) OR “canceration” (All Fields) OR “cancerization” (All Fields) OR “cancerized” (All Fields) OR “cancerous” (All Fields) OR “neoplasms” (MeSH Terms) OR “neoplasms” (All Fields) OR “cancer” (All Fields) OR “cancers” (All Fields).Translations, punica granatum: “pomegranate” (MeSH Terms) OR “pomegranate” (All Fields) OR (“punica” (All Fields) AND “granatum” (All Fields)) OR “punica granatum” (All Fields) cancer: “cancer’s” (All Fields) OR “cancerated” (All Fields) OR “canceration” (All Fields) OR “cancerization” (All Fields) OR “cancerized” (All Fields) OR “cancerous” (All Fields) OR “neoplasms” (MeSH Terms) OR “neoplasms” (All Fields) OR “cancer” (All Fields) OR “cancers” (All Fields).

### 2.5. Selection of Sources of Evidence

The search for eligible articles and reports was conducted by 2 reviewers (D.S. and M.D.) with a 3rd reviewer (G.A.C.) with the task of choosing whether to include the studies in situations of conflict.

The 2 reviewers after having established in agreement: the eligibility criteria, the keywords and the databases to be used, independently performed the search for articles and reports, reporting on tables the number of articles obtained for each keyword and for each database used; studies that resulted in duplicates from different databases were deleted using the EndNote 9 software (Philadelphia, PA, USA); study overlays that could not be uploaded to EndNote were manually removed by the authors after the screening phase. The 2 reviewers then proceeded to the screening and inclusion of the studies with the comparison and debate on the studies to be included.

### 2.6. Data Charting Process, Data Items, Synthesis of Results

The characteristics and type of data to be extracted from the studies were jointly decided by the 2 reviewers immediately after the study selection phase; the data concerned: the first author, the year of publication, the bibliographic reference, the type of study, the type of oral carcinoma investigated, the cell lines tested, the type of active ingredient tested, the main results and conclusions of the study. The data were extracted independently by the 2 reviewers in 2 different tables and subsequently compared and reported in a 3rd table with 3 reviewers who verified the correct insertion of the data.

### 2.7. Risk of Bias

The risk or bias was calculated using the Quality Assessment Tool For In Vitro Studies (QUIN Tool). The QUIN tool includes 12 points along with scoring and classification options to allow clinicians to assess the quality of in vitro studies; the tool has also been validated in dentistry, where results are presented by Sheth et al. [39].

For each criterion, a score was applied as follows: Adequately Specified (Score = 2), Inadequately Specified (Score = 1), Not Specified (Score = 0), Not Applicable.

The scores thus obtained were used to grade the in vitro study as high, medium or low risk (>70% = low risk of bias, 50% to 70% = medium risk of bias and <50% = high risk of bias) by using the following formula: Final score = Total score × 100\2 × number of criteria applicable.

The risk of bias was calculated by the first reviewer with a comparison with the 2nd reviewer in case of doubt.

Studies presenting high risk or bias were excluded from the scoping review.

## 3. Results

### 3.1. Results of Sources of Evidence

The search in: Science Direct Library, SCOPUS, PubMed and Cochrane produced a number of bibliographic sources equal to 1143. With the removal of duplicates, 885 potentially eligible articles were obtained, from 24 of which only 12 fully met the eligibility criteria but only 7 studies were included in the quantitative assessment.

Furthermore, the gray literature analysis (http://www.opengrey.eu DANS EASY Archive and Google Scholar) and previous systematic reviews did not allow for the identification of additional studies to be included in the quantitative assessment (Figure 1). The entire procedure of the identification, selection and inclusion of the studies is indicated in the flowchart of Figure 3.

### 3.2. Characteristics of Sources of Evidence, Results of Individual Sources of Evidence

A total of seven articles were included in the scoping review: Weisburg et al., 2010 [40], Morsy et al., 2019 [41], Peng et al., 2021 [42] Peng Et al., 2020, [43], Gao et al., 2022 [44], Seeram et al., 2005 [45]; Kasimsetty et al., 2010 [46]. All studies were in vitro (cell line studies). Two studies were conducted in Taiwan by the same research group, three studies were conducted in the USA, one study in Egypt and one in China. Three studies used POMx (POM Wonderful, LLC, Los Angeles, CA, USA) and three studies pomegranate juice (pomegranate polyphenol and tannin). The cell lines tested were as follows: Ca9-22 (OSCC, gingival squamous cell carcinoma); SCC9 (tongue squamous cell carcinoma); HSC-3 (tongue squamous cell carcinoma); HGF-1 (human gingival fibroblast); OC-2 (esophageal squamous cell carcinoma); Hep-2 (human papillomavirus-related endocervical adenocarcinoma); HSC-2 (oral cavity squamous cell carcinoma); CAL27 (tongue squamous cell carcinoma); SCC1483 (retromolar trigone squamous cell carcinoma); HF-1 (normal gingival fibroblasts); HSC-4 (tongue squamous cell carcinoma); BT-549 (ductal carcinoma, breast); HT-29 (colon carcinoma); KB (epidermal carcinoma, oral); SK-MEL (malignant melanoma); SKOV-3 (ovarian carcinoma); LLC-PK (Sus scrofa, kidney); Vero (Cercopithecus sabaeus, kidney); HCT116 (colon carcinoma); SW480 and SW620 (colon adenocarcinoma); RWPE-1 and 22Rv1 (prostate cancer) (Table 1).

In addition, the main oral cancer investigated was OSCC.

### 3.3. Quantitative Analysis

The quantitative analysis of the data extracted, reported in Table 2, did not foresee the execution of a meta-analysis of the data or an aggregation of the latter.

Given the high heterogeneity of the studies and data in question, it was therefore decided to report the most significant data on the inhibition of cell proliferation without performing further analyses.

However, all studies report an inhibitory effect on cell proliferation at variable concentrations (Table 2) for most of the oral cavity cancer cell lines.

### 3.4. Risk of Bias

All studies have an acceptable risk of bias, with a final score between 79% and 85%. However, all studies do not clearly report how sample randomization and operator blinding were achieved; this could be understood from the fact that these are in vitro studies on cell lines whose sample preparation may not be affected by the influence of sample randomization and operator blinding.

It was decided to apply a score of 1 to the quest after consulting with the second reviewer (Table 3)

## 4. Discussion

The authors performed a scoping review on the effects and potential uses of *Punica granatum* extracts on oral cancer; the review aims to summarize all the potential effects described in the literature. A preliminary analysis revealed the absence of studies conducted on patients with oral carcinoma or head and neck tumors; there are only in vitro studies on cell lines in which the main histological variant investigated was OSCC, as it emerges from the cell lines used (CAL27, SCC9, Ca9-22, HSC-3). The studies therefore identified and included in the review were seven in number.

From the analysis of the data in the literature, a clear inhibiting effect on cell proliferation in vitro emerges in almost all cell lines. In fact, the cytotoxic power was confirmed by a recent study by Gao et al., 2020, on four oral carcinoma cell lines (HSC-2, HSC-3, HSC-4 and Ca9-22) [44]. Moreover, the results on the reduction of cell proliferation are confirmed by the study of Seeram et al. 2005 [45], which indicates how the extracts of *Punica granatum* contain antioxidants capable of reducing the oxidative stresses that are one of the mechanisms responsible for mutations.

Specifically, different formulations can be identified with which the in vitro tests were performed: POMx (in capsule formulation containing powder), pomegranate juice extracts containing tannins and polyphenols and gold nanoparticles containing *Punica granatum* peel.

POMx is marketed both in the form of juice and powder concentrate formed from the peel, membrane and pith of pomegranate fruit, in which the main characteristic bioactive components are ellagitann ellagitannins (punicalagin and punicalin) and ellagic acid [47].

In the in vitro studies by Peng et al. [42] and Weisburg et al., 2010 [40], the powed formulation is used, which is obtained as described by Rasheed et al. [48], by extracting the pomegranate residue after pressing for juice and solid phase extraction to produce a powder with a high concentration of polyphenols. The concentration of substances would be the following: 86.0% ellagitannins, 2.5% ash, 3.2% sugars, 1.9% organic acids as citric acid equivalents, 0.8% nitrogen and 1.2% humidity. The percentage content of polyphenols was roughly divided into 19% ellagitannins such as punicalagin and punicalin, 4% free ellagic acid and 77% oligomers consisting of 2–10 repeated units of gallic acid, ellagic acid and glucose in different combinations [48].

As an alternative to the powder usually supplied in capsules, the second main formulation used is POM (Wonderful, LLC, Los Angeles, California) [45] or more generally pomegranate juice [46], or some main elagittannins (punicalagin) contained in the extracts as in Morsy et al., 2019 [41].

In the context of oral cavity tumors, the studies focused on two histological subtypes, OSCC and epidermal carcinoma of the mouth. In the first case, the cell lines used referred to tumors that were primarily localized in the gingival (Ca9- 22), tongue (CAL27, SCC9, HSC-3, HSC-4) and retromolar trigone area (SCC1483), or indicatively of the oral cavity (HSC-2); for the second histotype, the cell line used is the KB (the cell line may have contamination from the HeLa line, which appears to be cervical cancer) [49] (Figure 4).

The anticancer action of *Punica granatum* extracts is exerted through three main mechanisms: through the induction of apoptosis and through the inhibition of proliferation and invasion [50].

Oxidative stress is one of the mechanisms by which *Punica granatum* extracts can act by inducing apoptosis; the mechanism involves the generation of H_2_O_2_ with the depletion of glutathione; this mechanism has been observed and demonstrated for HSC2 oral cancer cell lines in the study of Weisburg et al., 2010 [40].

Therefore, the increase in ROS in oral cancer cells would be at the basis of apoptotic events, which also involve a reduction in the number of copies of mtDNA.

Moreover, there would also be an inhibiting effect on mitochondrial biogenesis with a reduction of the mitochondrial mass through the inhibition of the mRNA. Oral cancer cells treated with extracts of *Punica granatum* would undergo mitochondrial fission with consequent cellular apoptosis.

The anti-proliferative action against tumor cells is exerted through a generally cytotoxic effect performed at high concentrations with an apoptotic effect involving the inhibition of the phosphorylation of the MAPK family proteins such as JNK ERK1\2 and p38.

The antitumor action aimed at inhibiting tumor migration invasion is expressed through the suppression of the EMT process. In fact, Peng et al. demonstrate that EMT transcription factors (Slug and Twist) and mesenchymal markers (vimentin and N-cadherin) are downregulated [51], while the epithelial marker (E-cadherin) mRNA levels are upregulated in HSC-3 cell lines compared to control after 24 h of exposure to POMx [42,43].

Additional indirect anticancer mechanisms may be the antimicrobial activity detected by Kasimsetty et al., 2007 [52], with an antiplasmidial action presenting an inhibitory action also against candida albicans; these microorganisms are involved in some forms of oral pre-cancers [53] and the possibility of keeping them under control with a diet that includes the intake of *Punica granatum* extracts could be a prevention strategy. It should be added that the antioxidant capacity contained in the extracts of *Punica granatum* represents a determining factor in reducing the oxidative stress responsible for the mutagenic potential of cancer cells, mutations potentially involving tumor suppressor genes such as p53 [54] (Figure 5).

A strong limitation to this purpose review is the absence of clinical studies conducted on oral carcinomas; however, the risk of publication bias has been minimized through research on gray literature sources [55].

## 5. Conclusions

In conclusion, we can say that *Punica granatum* extracts represent an antioxidant nutraceutical food source that is important from an antitumor point of view, also considering the properties that inhibit proliferation and invasion and the ability to induce apoptosis on many tumors cell lines.

Even if there are only preclinical in vitro studies, the preliminary data on the cellular lines of OSCC indicate that for oral cavity tumors there are conditions for which the extracts of Punic granatum prove to be effective at least from a perspective of prevention.

## Figures and Tables

**Figure 1 dentistry-10-00234-f001:**
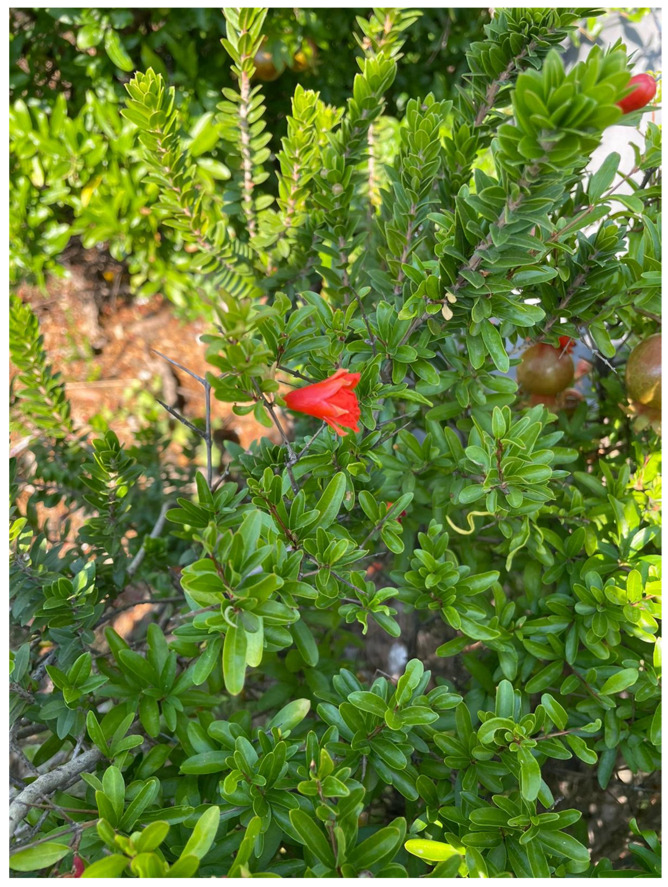
Photo of a *Punica granatum* in which the inflorescences are visible and the presence of pomegranate fruits at different stages of development. Photographs taken by Diego Sovereto in Trieste (Italy) in July 2022.

**Figure 2 dentistry-10-00234-f002:**
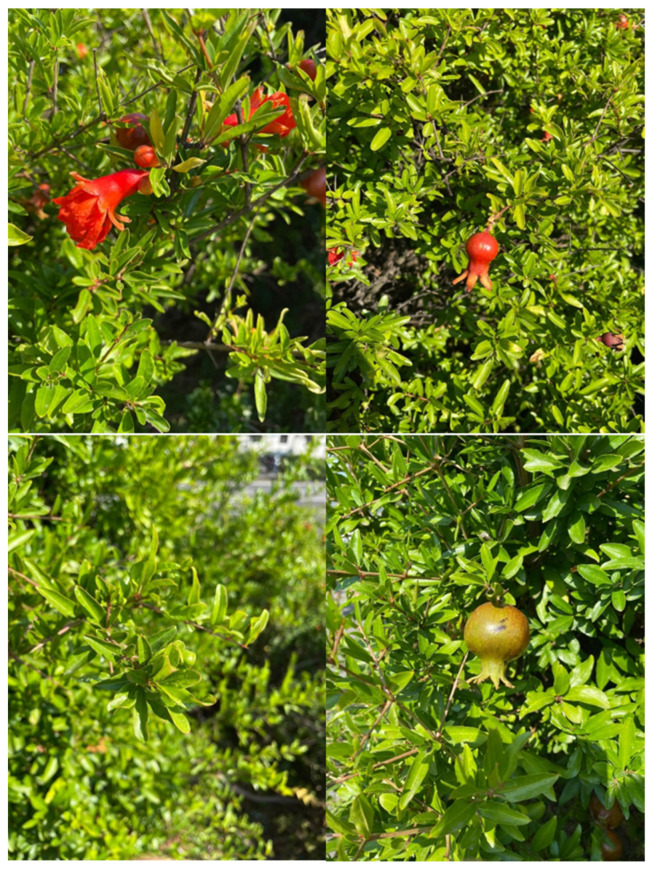
Photo of *Punica granatum* with particularities of leaves, fruits and inflorescences. Photographs taken by Diego Sovereto in Trieste (Italy) in July 2022.

**Figure 3 dentistry-10-00234-f003:**
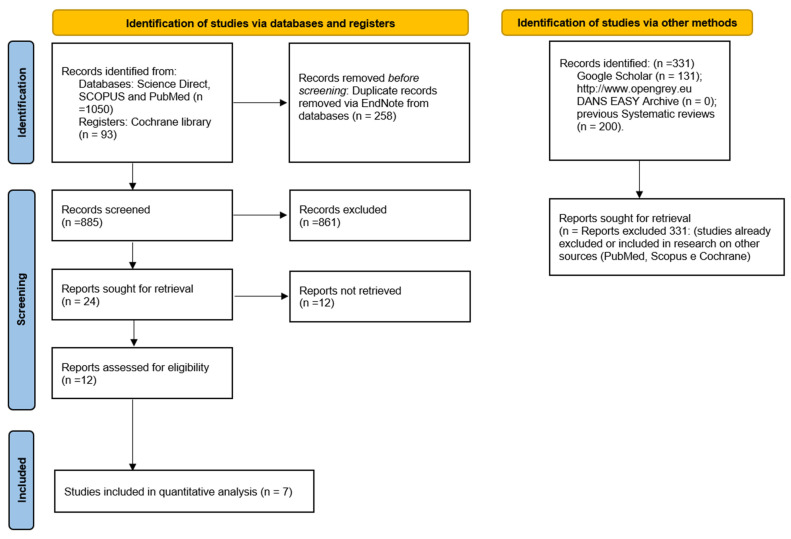
Entire selection and screening procedures are described in the PRISMA flowchart.

**Figure 4 dentistry-10-00234-f004:**
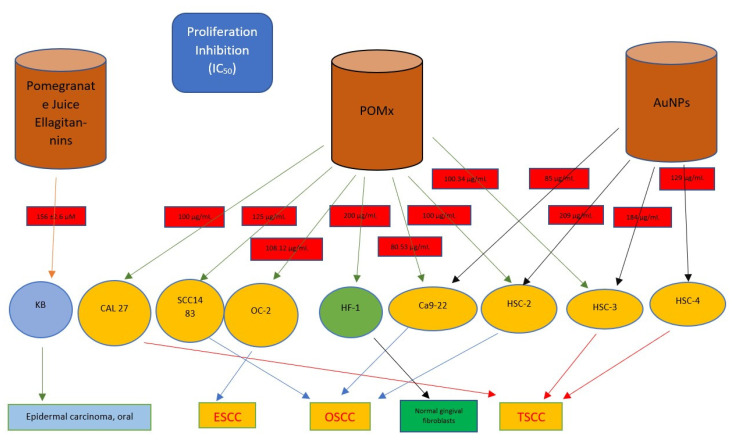
The IC_50_ concentration values of the three main products deriving from *Punica granatum* studied in the literature against the main carcinomatous cell lines of the oral cavity are shown in the graph; ESCC (Esophageal Squamous Cell Carcinoma), TSCC (Tongue Squamous Cell Carcinoma).

**Figure 5 dentistry-10-00234-f005:**
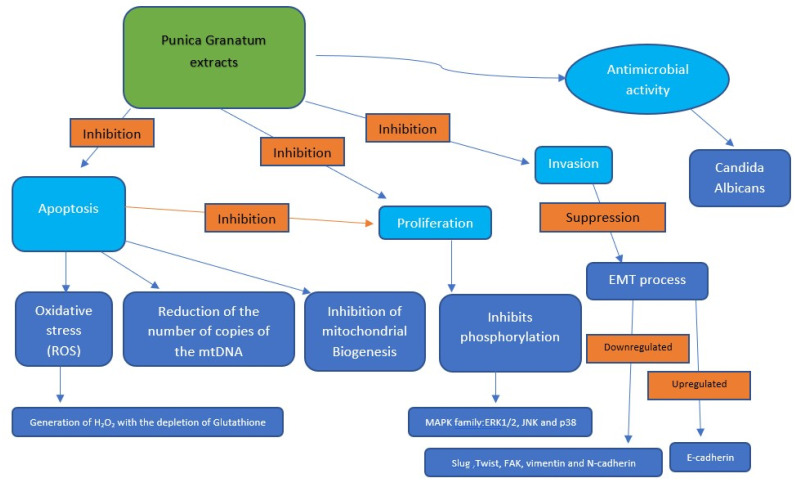
The main mechanisms by which *Punica granatum* extracts carry out their potential antitumor activity are represented in the graph; the mechanisms are direct and indirect. The direct mechanisms foresee an antiapoptotic action carried out through oxidative stress with the production of ROS also acting on the cellular proliferative capacity. The antiproliferative activity is exerted not only with apoptosis but also with inhibition of the phosphorylation of proteins mainly of the MAPK family; finally, we have as a direct mechanism an inhibition of the invasion by inhibition of the EMT (epithelial-mesenchymal transition) process. The indirect mechanisms involve an antibacterial and antifungal activity carried out by the action of the *Punica granatum* extracts; the antifungal action against Candida albicans is fairly documented in the literature. The inhibition of the activity of these microorganisms reduces the inflammatory picture induced by the invasion with a reduction of the risk of oral tissue cancer.

**Table 1 dentistry-10-00234-t001:** In vitro studies: the main characteristics of the included studies are reported with the main results.

First Author, Data	Country	Type of Study	Pathologies	Cell Lines	*Punica granatum* L. Extract	Main Findings of the Study
Peng Et al., 2020 [43]	Taiwan	vitro	OSCC,	SCC9, Ca9-22, HSC-3, HGF-1	POMx pomegranate powder (POM Wonderful, LLC, Los Angeles, CA, USA),	low cytotoxic concentrations of POMx inhibited cell migration and invasion of oral cancer cells
Peng et al., 2021 [42]	Taiwan	vitro	OSCC, esophageal squamous cell carcinoma	Ca9-22, HSC-3, OC-2	POMx pomegranate powder	POMx provides antiproliferative and apoptotic effects on oral cancer cells due to impaired mitochondrial functioning
Morsy et al., 2019 [41]	Egypt	vitro	endocervical adenocarcinoma	Hep-2	Punicalagin (pomegranate polyphenol)	Pomegranate can be an adjuvant, natural product in oral cancer treatment through its anti-proliferative, anti-angiogenic as well as apoptotic activity
Weisburg et al., 2010 [40]	USA	vitro	Follicular lymphoma, OSCC	HSC-2 cells CAL27, SCC1483, HF-1	POMx	Pomegranate extract may have the potential as a chemopreventive agent for the oral cavity
Gao et al., 2022 [44]	China	vitro	OSCC	HSC-2, HSC-3, HSC-4, Ca9-22	AuNPs (gold nanoparticles) containing *Punica granatum* peel)	The best result of cytotoxicity property of AuNPs against the above cell lines was seen in the case of the HSC-3 cell line
Seeram et al., 2005 [45]	USA	vitro	epidermal carcinoma of the mouth, OSCC, colon carcinoma, prostate cancer	KB, CAL27, HT-29, HCT116, SW480, SW620, RWPE-1, 22Rv1	POM pomegranate juice (POM Wonderful, LLC, Los Angeles, CA, USA), (tannin, punicalagin, ellagic acid and polyphenols)	Pomegranate juice decreased the viability of cancer cell lines of the oral cavity, prostate and colon
Kasimsetty et al., 2010 [46]	USA	vitro	colon carcinoma, malignant melanoma, epidermal carcinoma of the mouth, ductal carcinoma breast, ovarian carcinoma	BT-549, HT-29, KB, SK-MEL, SKOV-3, LLC-PK, Vero	Pomegranate Juice Ellagitannins	The results indicate that the consumption of pomegranate juice in considerable quantities could potentially reduce the risk of developing colon cancer by inhibiting cell proliferation and inducing apoptosis.

**Table 2 dentistry-10-00234-t002:** Quantitative evaluation of the data extracted from the studies: in particular, the values concerning the concentrations that have obtained an inhibitory value of the proliferation of the various cell lines of the oral cavity are reported.

	Cell Viability	Cell Lines of the Oral Cavity									Main Types of Substances and Extracts of Pomegranate Tested
First author, data		SCC1483	Ca9-22	CAL 27	HSC-2	HSC-3	HSC-4	OC-2	HF-1 fibroblasts	KB	
Peng et al., 2021 [42]	IC50 value at 24 h	\	80.53 µg/mL	\	\	100.34 µg/mL	\	108.12 µg/mL	\	\	POMx
Weisburg et al., 2010 [40]	Midpoint cytotoxicity (NR50) values at 24 h	125 µg/mL	\	100 µg/mL	100 µg/mL	\	\	\	200 µg/mL	\	POMx
Gao et al., 2022 [44]	IC50 value at 48 h	\	85 µg/mL	\	209 µg/mL	184 µg/mL	129 µg/mL	\	\	\	AuNPs
Kasimsetty et al., 2010 [46]	IC50 value at 48 h	\	\	\	\	\		\	\	156 ± 2.6 μM	Pomegranate Juice Ellagitan-nins
Peng Et al., 2020 [43]	POMx treatment under 25 and 50 μg/mL shows >80% cell viability in four oral cancer cells (HSC-3, Ca9-22, SCC9 and OC-2) after 24 h										POMx
Seeram et al., 2005 [45]	Antiproliferative activity against human oral (KB, CAL27): PJ showed the greatest antiproliferative activity against all cell lines by inhibiting proliferation from 30% to 100%										punicalagin, ellagic acid and a total pomegranate tannin extract
Morsy et al., 2019, [41]	Punicalagin was used in a concentration of (20 µg mLG1) for 24 h and had a synergistic anticancer effect on Hep-2 cell line when used in conjunction with either a high or low dose of 5-fluorouracil (5-FU)										Punicalagins

**Table 3 dentistry-10-00234-t003:** QUIN Tool (quality assessment tool for in vitro studies): For each criterion a score will be applied as follows: Adequately Specified (Score = 2), Inadequately Specified (Score = 1), Not Specified (Score = 0), Not Applicable.

First Author, Data	Clearly Stated Aims/Objectives	Detailed Explanation of Sample Size Calculation	Detailed Explanation of Sampling Technique	Details of Comparison Group	Detailed Explanation of Methodology	Operator Details	Randomization	Method of Measurement of Outcome	Outcome Assessor Details	Blinding	Statistical Analysis	Presentation of Results
Peng Et al., 2020 [43]	2	1	2	2	2	1	1	2	2	1	2	2
Peng et al., 2021 [42]	2	1	2	2	2	1	1	2	2	1	2	2
Morsy et al., 2019 [41]	2	1	2	2	2	1	1	2	2	1	2	1
Weisburg et al., 2010 [40]	2	2	2	2	2	1	1	2	2	1	2	2
Gao et al., 2022 [44]	2	2	2	2	2	1	1	2	2	1	2	2
Seeram et al., 2005 [45]	2	1	2	2	2	1	1	2	2	1	2	2
Kasimsetty et al., 2010 [46]	2	1	2	2	2	1	1	2	2	1	2	2

## Data Availability

Not applicable.

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
