# Peer review of "Application of the Extracts of Punica granatum in Oral Cancer: Scoping Review"

_dentistry, 2022, doi:10.3390/dj10120234_

Round 1

Reviewer 1 Report

Dear Authors,

Congratulations on your work! I consider that some aspects need to be added:

the quantitative results of the studies included in the review

the risk of bias in the selected studies

Author Response

Reviewer 1

Dear Authors,

Congratulations on your work! I consider that some aspects need to be added:

the quantitative results of the studies included in the review

the risk of bias in the selected studies

ANSWER

Thanks for the suggestions and advice given to me; they have proved to be very useful for improving the manuscript.

  • A quantitative evaluation of the extracted results was performed and the data were reported in a new table as follows.

3. Analisy quantitative.

The quantitative analysis of the data extracted, reported in table 2, did not foresee the execution of a meta-analysis of the data or an aggregation of the latter.

Given the high heterogeneity of the studies and data in question, it was therefore decided to report the most significant data on the inhibition of cell proliferation without performing further analyses.

However, all studies report an inhibitory effect on cell proliferation at variable concentrations (Table 2) for most of the oral cavity cancer cell lines.

Table.2 quantitative evaluation of the data extracted from the studies: in particular, the values concerning the concentrations that have obtained an inhibitory value of the proliferation of the various cell lines of the oral cavity are reported.

Cell viability

Cell lines of the oral cavity

Main types of substances and extracts of pomegranate tested

First autor, data

SCC1483

Ca9-22

CAL 27

HSC-2

HSC-3

HSC-4

OC-2

HF-1 fibroblasts

KB

Peng et al. 2021,[42]

IC50 value at

24 h

\

80.53 µg/mL

\

\

100.34  µg/mL

\

108.12 µg/mL

\

\

 POMx

Weisburg et al. 2010, [40]

Midpoint cytotoxicity

(NR50) values at 24 H

125 µg/mL

\

100 µg/mL

100 µg/mL

\

\

\

200 µg/mL 

\

POMx

 Gao et al. 2022 [44]

IC50 value at 48 H

\

85 µg/mL

\

209 µg/mL

184 µg/mL

129 µg/mL

\

\

\

AuNPs

Kasimsetty et al. 2010 [46]

IC50 value at 48 H

\

\

\

\

\

\

\

156 ±2.6 μM

Pomegranate Juice Ellagitan-nins

Peng Et al. 2020,[43]

POMx treatment under 25 and 50 μg/mL shows >80% cell viability in four oral cancer cells (HSC-3, Ca9-22, SCC9, and OC-2) after 24 hours

 POMx

Seeram et al. 2005[45]

antiproliferative activity against human oral (KB, CAL27):  PJ showed the greatest antiproliferative activity

against all cell lines by inhibiting proliferation from 30% to 100%

punicalagin, ellagic acid and a total pomegranate tannin extract

Morsy et al. 2019, [41]

 Punicalagin was used in a concentration of (20 µg mLG1) for 24 h and have a synergistic anticancer effect on Hep-2 cell line when used in conjunction with either a high or low dose of 5-fluorouracil (5-FU)

Punicalagins

  • A risk of bias assessment of included studies was performed:

3.3  Risk of Bias

All studies have an acceptable risk of bias, with a final score between 79%

and 85%. However, all studies do not clearly report how sample randomization and operator blinding were achieved; This could be understood from the fact that these are in vitro studies on cell lines whose sample preparation may not be affected by the influence of sample randomization and operator blinding.

 It was decided to apply a score of 1 to the quest after consulting with the second reviewer (Table 3)

 Table.3 QUIN Tool (quality assessment tool for in vitro studies):For each criterion a score will be applied as follows: Adequately Specified (Score=2),Inadequately Specified (Score=1) ;Not Specified (Score=0),Not Applicable;

First autor, data

Clearly stated aims/objectives

Detailed explanation of sample size calculation

Detailed explanation of sampling technique

Details of comparison group

Detailed explanation of methodology

Operator details

Randomization

Method of measurement of outcome

Outcome assessor details

Blinding

Statistical analysis

Presentation of results

Peng Et al. 2020,[43]

2

1

2

2

2

1

1

2

2

1

2

2

Peng et al. 2021,[42]

2

1

2

2

2

1

1

2

2

1

2

2

Morsy et al. 2019, [41]

2

1

2

2

2

1

1

2

2

1

2

1

Weisburg et al. 2010, [40]

2

2

2

2

2

1

1

2

2

1

2

2

 Gao et al. 2022 [44]

2

2

2

2

2

1

1

2

2

1

2

2

Seeram  et al. 2005[45]

2

1

2

2

2

1

1

2

2

1

2

2

Kasimsetty et al. 2010 [46]

2

1

2

2

2

1

1

2

2

1

2

2

Best regards Mario Dioguardi

Reviewer 2 Report

Dear authors

The article is well written and the work has scientific relevance.

I have some suggestions/questions.

-On page 6 (line 183) start the paragraph with a capital letter;

- In "2.3. information sources", start with a capital letter, as are the other points;

- At the beginning of the conclusion, fix: "n conclusion, we can...";

- There are two "figure 1" in the article. Fix it.

Author Response

Reviewer 2

Dear authors

The article is well written and the work has scientific relevance.

I have some suggestions/questions.

-On page 6 (line 183) start the paragraph with a capital letter;

- In "2.3. information sources", start with a capital letter, as are the other points;

- At the beginning of the conclusion, fix: "n conclusion, we can...";

- There are two "figure 1" in the article. Fix it.

ANSWER

Thank you for reviewing the manuscript. All corrections have been made as requested

 Best regards Mario Dioguardi

Reviewer 3 Report

In this manuscript, Mario Dioguardi and Andrea Ballini described the application of the extracts of Punica Granatum in oral cancer. In this review, they summarized this conclusion that from the preliminary data on the cellular lines of OSCC it emerges that for oral cancer there are the conditions for which the extracts of Punic granatum are effective at least from a prevention perspective.

The strengths of this article are as followed: Punica Granatum is widely used as food or medicine, and it is widely cultivated and loved by people. This article makes a comprehensive statistics of the application and research on its anti-tumor effect published in recent years. The article still needs to be revised as followed:   

The disadvantages of this article are as followed: This article is just a simple summary and listing of several articles, and does not condense and refine the key points of this article. At the same time, there are not many charts to support and describe its conclusions. It is suggested to add appropriate content.

Author Response

Reviewer 3

In this manuscript, Mario Dioguardi and Andrea Ballini described the application of the extracts of Punica Granatum in oral cancer. In this review, they summarized this conclusion that from the preliminary data on the cellular lines of OSCC it emerges that for oral cancer there are the conditions for which the extracts of Punic granatum are effective at least from a prevention perspective.

The strengths of this article are as followed: Punica Granatum is widely used as food or medicine, and it is widely cultivated and loved by people. This article makes a comprehensive statistics of the application and research on its anti-tumor effect published in recent years. The article still needs to be revised as followed:   

The disadvantages of this article are as followed: This article is just a simple summary and listing of several articles, and does not condense and refine the key points of this article. At the same time, there are not many charts to support and describe its conclusions. It is suggested to add appropriate content.

Answer

Thank you for reviewing the manuscript, your advice has been very helpful in improving the manuscript. In fact, as suggested, I have added charts to support my conclusions under discussion with the aim of summarizing the key points of the discussion and of the extracted data.

 Furthermore I have added a quantitative analysis of the data extracted on the inhibitory activity of Punica granatume extracts and I have performed an evaluation of the risk of Bias. all added parts are shown in yellow in the text .

3.3. Analisy quantitative.

The quantitative analysis of the data extracted, reported in table 2, did not foresee the execution of a meta-analysis of the data or an aggregation of the latter.

Given the high heterogeneity of the studies and data in question, it was therefore decided to report the most significant data on the inhibition of cell proliferation without performing further analyses.

However, all studies report an inhibitory effect on cell proliferation at variable concentrations (Table 2) for most of the oral cavity cancer cell lines.

Table.2 quantitative evaluation of the data extracted from the studies: in particular, the values concerning the concentrations that have obtained an inhibitory value of the proliferation of the various cell lines of the oral cavity are reported.

Cell viability

Cell lines of the oral cavity

Main types of substances and extracts of pomegranate tested

First autor, data

SCC1483

Ca9-22

CAL 27

HSC-2

HSC-3

HSC-4

OC-2

HF-1 fibroblasts

KB

Peng et al. 2021,[42]

IC50 value at

24 h

\

80.53 µg/mL

\

\

100.34  µg/mL

\

108.12 µg/mL

\

\

 POMx

Weisburg et al. 2010, [40]

Midpoint cytotoxicity

(NR50) values at 24 H

125 µg/mL

\

100 µg/mL

100 µg/mL

\

\

\

200 µg/mL 

\

POMx

 Gao et al. 2022 [44]

IC50 value at 48 H

\

85 µg/mL

\

209 µg/mL

184 µg/mL

129 µg/mL

\

\

\

AuNPs

Kasimsetty et al. 2010 [46]

IC50 value at 48 H

\

\

\

\

\

\

\

156 ±2.6 μM

Pomegranate Juice Ellagitan-nins

Peng Et al. 2020,[43]

POMx treatment under 25 and 50 μg/mL shows >80% cell viability in four oral cancer cells (HSC-3, Ca9-22, SCC9, and OC-2) after 24 hours

 POMx

Seeram et al. 2005[45]

antiproliferative activity against human oral (KB, CAL27):  PJ showed the greatest antiproliferative activity

against all cell lines by inhibiting proliferation from 30% to 100%

punicalagin, ellagic acid and a total pomegranate tannin extract

Morsy et al. 2019, [41]

 Punicalagin was used in a concentration of (20 µg mLG1) for 24 h and have a synergistic anticancer effect on Hep-2 cell line when used in conjunction with either a high or low dose of 5-fluorouracil (5-FU)

Punicalagins

3.3  Risk of Bias

All studies have an acceptable risk of bias, with a final score between 79% and 85%. However, all studies do not clearly report how sample randomization and operator blinding were achieved; This could be understood from the fact that these are in vitro studies on cell lines whose sample preparation may not be affected by the influence of sample randomization and operator blinding.

 It was decided to apply a score of 1 to the quest after consulting with the second reviewer (Table 3)

 Table.3 QUIN Tool (quality assessment tool for in vitro studies):For each criterion a score will be applied as follows: Adequately Specified (Score=2),Inadequately Specified (Score=1) ;Not Specified (Score=0),Not Applicable;

First autor, data

Clearly stated aims/objectives

Detailed explanation of sample size calculation

Detailed explanation of sampling technique

Details of comparison group

Detailed explanation of methodology

Operator details

Randomization

Method of measurement of outcome

Outcome assessor details

Blinding

Statistical analysis

Presentation of results

Peng Et al. 2020,[43]

2

1

2

2

2

1

1

2

2

1

2

2

Peng et al. 2021,[42]

2

1

2

2

2

1

1

2

2

1

2

2

Morsy et al. 2019, [41]

2

1

2

2

2

1

1

2

2

1

2

1

Weisburg et al. 2010, [40]

2

2

2

2

2

1

1

2

2

1

2

2

 Gao et al. 2022 [44]

2

2

2

2

2

1

1

2

2

1

2

2

Seeram  et al. 2005[45]

2

1

2

2

2

1

1

2

2

1

2

2

Kasimsetty et al. 2010 [46]

2

1

2

2

2

1

1

2

2

1

2

2

……………………………………………………………………………………………………………………………………………………………………………………………

In the context of oral cavity tumors, the studies focused on 2 histological subtypes, OSCC and epidermal carcinoma of the mouth, in the first case the cell lines used referred to tumors that were primarily localized on the gingival (Ca9- 22), tongue (CAL27, SCC9, HSC-3, HSC-4) retromolar trigone area (SCC1483), or indicatively of the oral cavity (HSC-2), for the second histotype the cell line used is the KB (The cell line may have contamination from the HeLa line which appears to be cervical cancer)[49] (Figure 4).

Figure 4. The IC50 concentration values of the 3 main products deriving from Punica granatum studied in the literature against the main carcinomatous cell lines of the oral cavity are shown in the graph; ESCC (Esophageal Squamous Cell Carcinoma), TSCC (Tongue Squamous Cell Carcinoma)

The anticancer action of Punica granatum extracts is exerted through 3 main mechanisms, through the induction of apoptosis and through the inhibition of proliferation and invasion [50].

Oxidative stress is one of the mechanisms by which Punica granatum extracts can act by inducing apoptosis, the mechanism involves the generation of H2O2 with the depletion of Glutathione, this mechanism has been observed and demonstrated for HSC2 oral cancer cell lines. in the study by Weisburg et al. 2010 [40].

Therefore, the increase of ROS in oral cancer cells would be at the basis of apoptotic events which also involve a reduction in the number of copies of mtDNA.

Moreover, there would also be an inhibiting effect on mitochondrial biogenesis with reduction of the mitochondrial mass through the inhibition of the mRNA. Oral cancer cells treated with extracts of Punica granatum would undergo mitochondrial fission with consequent cellular apoptosis.

The anti-proliferative action against tumor cells is exerted through a generally cytotoxic effect performed at high concentrations with an apoptotic effect, to be involved would be the inhibition of the phosphorylation of the MAPK family proteins such as JNK ERK1\2 and p38.

The antitumor action aimed at inhibiting tumor migration invasion is expressed through the suppression of the EMT process. In fact, Peng et al.  demonstrates that EMT transcription factors (Slug and Twist) and mesenchymal markers (vimentin and N-cadherin) are downregulated [51], while the Epithelial marker (E-cadherin) mRNA levels are upregulated in HSC-3 cell lines compared to control after 24 hours of exposure to POMx [42,43].

Additional indirect anticancer mechanisms may be the antimicrobial activity detected by Kasimsetty et al. 2007 [52] with an antiplasmidial action presented an inhibitory action also against candida albicans, these microorganisms are involved in some forms of oral precancerous [53]  and the possibility of keeping them under control with a diet that includes the intake of Punica granatum extracts could be a prevention strategy. It should be added that the antioxidant capacity contained in the extracts of Punica granatum represents a determining factor in reducing the oxidative stress responsible for the mutagenic potential of cancer cells; Mutations potentially involving tumor suppressor genes such as p53 [54] (Figure 5).

Figure 5. The main mechanisms by which Punica granatum extracts carry out their potential antitumor activity are represented in the graph: the mechanisms are direct and indirect.

The direct mechanisms foresee an antiapoptotic action carried out through an oxidative stress with the production of ROS, also acting on the cellular proliferative capacity. The antiproliferative activity is exerted not only with apoptosis but also with inhibition of the phosphorylation of proteins mainly of the MAPK family; finally, we have as a direct mechanism an inhibition of the invasion by inhibition of the EMT (epithelial-mesenchymal transition) process.

The indirect mechanisms involve an antibacterial and antifungal activity carried out by the action of the Punica granatum extracts; the antifungal action against Candida albicans is fairly documented in the literature. The inhibition of the activity of these microorganisms reduces the inflammatory picture induced by the invasion with a reduction of the risk of oral tissue cancer.

Round 2

Reviewer 3 Report

This paper has been revised based on the suggestions put forward last time, which basically meet the requirements of this journal, and it is recommended to accept.